# Socioeconomic trajectories of body mass index and waist circumference: results from the English Longitudinal Study of Ageing

Paola Zaninotto, Camille Lassale

Epidemiology and Public Health, University College London, London, UK

**Correspondence to**
Dr Paola Zaninotto;
p.zaninotto@ucl.ac.uk

## ABSTRACT

**Objectives** To explore age trajectories of body mass index (BMI) and waist circumference (WC) and to examine whether these trajectories varied by wealth.

**Design** Nationally representative prospective cohort study.

**Setting** Observational study of people living in England.

**Participants** 7416 participants aged 52 and over of the English Longitudinal Study of Ageing (2004–2012).

**Primary outcome measures** BMI and WC assessed objectively by a trained nurse.

**Main exposure measure** Total non-pension household wealth quintiles defined as financial wealth, physical wealth (such as business wealth, land or jewels) and housing wealth (primary and secondary residential housing wealth), minus debts.

**Results** Using latent growth curve models, we showed that BMI increased by 0.03 kg/m$^2$ (95% CI 0.02 to 0.04, p<0.001) per year and WC by 0.18 cm (95% CI 0.15 to 0.22, p<0.001). Age (linear and quadratic) showed a negative association with BMI and WC baseline and rates of change, indicating that older individuals had smaller body sizes and that the positive rates of change flattened to eventually become negative. The decline occurred around the age of 71 years for BMI and 80 years for WC. Poorest wealth was significantly related to higher baseline levels of BMI (1.97 kg/m$^2$ 95% CI 0.99 to 1.55, p<0.001) and WC (4.66 cm 95% CI 3.68 to 2.40, p<0.001). However, no significant difference was found in the rate of change of BMI and WC by wealth, meaning that the age trajectories of BMI and WC were parallel across wealth categories and that the socioeconomic gap did not close at older ages.

**Conclusions** Older English adults showed an increase in BMI and WC over time but this trend reversed at older old age to display a sharp decrease. At any given age wealthier people had more favourable BMI and WC profile.

## Strengths and limitations of this study

► This is the first large nationally representative study in England to depict age-specific trajectories of body mass index and waist circumference over an 8-year period and to examine whether these differed by wealth.
► A trained nurse assessed body mass index and waist circumference objectively.
► This study employed wealth as measure of socioeconomic status that is particularly relevant at older ages, since it reflects both past and current socioeconomic status.
► The study findings are limited to men and women aged 52 and over.

## INTRODUCTION

There has been a marked increase in the number of obese older adults over the past decades, and the number is set to grow as the population continues to age. In the UK, it is estimated that there will be an additional 11 million obese adults by 2030, of whom 3.3 million aged over 60 years.[1] The obesity epidemic has negative health consequences also in older adults where excess adiposity has been associated with chronic diseases,[2 3] disability[4 5] and mortality.[6 7]

It is increasingly recognised that a dynamic measure of body mass index (BMI) better predicts future health risks among older adults.[8 9] In older adults it has been shown that BMI increases over time up to early old age,[10 11] and declines thereafter[12 13] due to loss in muscle mass (sarcopenia).

Growing evidence indicates differential cardiovascular risk according to adipose depots location, with higher risk observed with visceral compared with subcutaneous fat.[14]

Therefore, in older adults waist circumference (WC) can be a better predictor of cardiovascular risk than BMI, and both indicators are important to monitor total and visceral adiposity.

Total and central obesity are socially patterned in developed countries with higher rates of obesity observed in lower socioeconomic groups.[15 16] However, there is limited evidence on BMI trajectories in relation to socioeconomic indicators in older adults, and results are contradictory.[10 11 17] To our

knowledge, no studies from developed countries specifically reported socioeconomic trajectories of WC among older adults. Furthermore, despite evidence of the importance of using wealth as an indicator of socioeconomic status of older individuals,[18–20] no studies have investigated longitudinal changes in BMI and WC among older adults by wealth.

Our study has two specific aims. The first objective is to depict age trajectories of BMI and WC by analysing 8 years (2004–2005 to 2012–2013) of longitudinal data from a nationally representative sample of English adults aged 52 and over. The second objective is to examine whether these trajectories vary by wealth. To allow reliable tracking of changes over time, we use growth curve models which enable studying between-person differences in within-person change by fitting fixed and random effects.[21]

## METHODS

The data are from three waves of the English Longitudinal Study of Ageing (ELSA),[22] where the same individuals, aged 50 and older and living in private households in England, were followed and reinterviewed every 2 years. The study began in 2002–2003 (first phase of data collection referred to as wave 1) with 11 391 individuals. Data collection includes a face-to-face interview and a self-completion questionnaire at each wave and trained nurses performed visits in the participants' homes at alternative waves, to collect anthropometry measures and blood samples. For the purpose of this study, we used data from wave 2 (2004–2005), wave 4 (2008–2009) and wave 6 (2012–2013) where nurse visits took place and anthropometric measurements were taken. The sample size consisted of 7255 (5042 at wave 4 and 4256 at wave 6) for the study of BMI trajectories and 7416 (4907 at wave4 and 4147 at wave 6) for the study of WC trajectories. Attrition rates are presented in online supplementary table S1. Total follow-up time was 8 years (average 4).

### Patient and public involvement statement

Patients did not participate in the design and data analysis of this study.

### Anthropometric measurements

Height, weight and WC were measured during the nurse visit. Height without shoes was measured using a portable stadiometer. One measurement was taken with the informants stretching to the maximum height and the head in the Frankfort plane. Weight was measured using a portable electronic scale. Informants were asked to remove their shoes and any bulky clothing. BMI was defined as weight (kg)/height (m$^2$) (ranged from 14.9 to 56.2 kg/m$^2$). WC was defined as the midpoint between the lower rib and the upper margin of the iliac crest. Two measurements were taken using a tape with an insertion buckle at one end. WC was recorded to the nearest even millimetre. The mean values of the two valid measurements (the two that were the closest to each other, if

there were three measurements) were used in the analysis (ranged between 60.8 and 108.0 cm).

### Wealth

Total non-pension household wealth, reported at the household level, was defined as financial wealth, physical wealth (such as business wealth, land or jewels) and housing wealth (primary and secondary residential housing wealth), minus debts. The variable is excluding regular pension payments, but includes lump sums from private pension that have already been received but not yet consumed. The total score (range −126 990 to 9 319 227, mean 276 702 SD 396 453) was divided into five equal quintiles. More detailed information can be found in online supplementary materials.

### Covariates

We used a continuous variable for age (ranging from 52 to 101 at baseline) and a dichotomous variable for sex (0"male' 1"female'). From information on marital and cohabiting status, we derived a dichotomous variable for cohabitation status (0"Currently living with a partner whether married or not' 1"Currently not living with a partner'). During the interview, respondents were asked whether they had any long-standing illnesses, and if the illnesses limited their daily activities, responses were combined into a dichotomous variable, classifying participants as having a limiting long-standing illness or not.

### Statistical analysis

To best depict reliable trajectories of anthropometric measures and to provide insight into how the rate of change relates to other variables, we used latent growth curve modelling. Growth curve modelling is specifically designed to capture change over time, by estimating latent growth factors for the baseline status (intercept) and rate of change (linear slope) over the 8-year period.[21 23–26] Latent factors representing intercept and slope components were derived from the three observations of anthropometric measures at wave 2 (baseline), wave 4 and wave 6. Factor loadings of the intercept component to all three observations were fixed to 1, and the linear slope component was defined by fixing the parameters to 0 (baseline, wave 2), 4 (wave 4) and 8 (wave 6) so that the slope parameter can be interpreted as the change per year. To depict age trajectories of BMI and WC, we fitted models with a linear and a quadratic term of age (non-adjusted). To answer our second research question, we tested the associations of wealth quintiles (treated as categorical in the models with 'richest wealth' as the reference category) with intercept and slope. All models included age (linear and quadratic, centred at the mean (66 years)), sex (treated as binary), cohabiting status (treated as binary, centred at the mean (0.51)), limiting long-standing illness (treated as binary, centred at the mean (0.67)) as covariates, all measured at baseline. To ease interpretation of estimates for age (linear and quadratic), we used a 10-year increment in age, instead of

single years. For simplicity, we only present parameter estimates for age, sex and wealth. To determine the fit of the models, we employed the Comparative Fit Index (CFI), the Tucker-Lewis Index (TLI) and the root-mean-square error of approximation (RMSEA), which represents closeness of fit.[27] The growth curve models (and descriptive statistics) accounted for the complex survey design and were weighted to adjust for non-response and to make the ELSA sample representative of the population of adults aged 50 and over, living in private households in England. Missing data were handled using full information maximum likelihood estimation, which computes parameter estimates on the basis of all available data under the assumption that data are missing at random. Models were fitted using Mplus V.7.

In order to show graphically the level of BMI and WC at baseline, direction and amount of change throughout the age range of our sample, we present ageing-vector graphs[28] of predicted BMI and WC. The ageing graphs were fitted using Stata V.15.

## Sensitivity analyses

We ran a sensitivity analysis to test whether attrition due to mortality or drop-out violated the missing at random assumption. To test for this, we jointly modelled loss to follow-up with the latent growth curve. Logistic regression was used to estimate the probability of loss to follow-up at wave t based on the individual's age and wealth at baseline and BMI or WC at $t-1$.

We also tested for interaction terms between wealth and sex for each outcome, BMI and WC, as the impact of wealth on body size trajectory may differ by sex.

## RESULTS

Baseline (2004–2005) descriptive characteristics of the samples are shown in table 1. In the 'BMI' analytical sample (n=7225), overall mean age at baseline was 66, and average BMI was 27.9 kg/m², 53% of people were women; 70% were cohabiting with a partner and 34% reported having a limiting long-standing illness. The 'WC' analytical sample (n=7416) was essentially similar and the average WC was 95.8 cm.

## BMI and WC age trajectories

Table 2 shows the baseline level of BMI (28.19 kg/m²) and WC (96.98 cm) and change overtime (BMI slope 0.030 kg/m² and WC slope 0.183 cm) after accounting for age. The intercept value refers to the average value of members that were aged 66 at baseline, and the slope refers to the increase (in kg/m² for BMI or in cm for WC) per each additional year of the study. Age was negatively associated both with intercept and slope. The negative estimate of age on the intercept reflects a cohort effect, whereby older generations have lower values of BMI and WC than younger generations. The negative effect of age on the slope means that the overall positive slope observed at age 66 is actually negative at older ages. The ageing-vector graphs shown in figure 1 provide a visual summary. Each arrow (vector) represents the predicted BMI and WC at baseline and the 8-year change for every 2-year age cohort. The arrows initiate at the beginning of the study period (wave 2, 2004–2005) and finish at the end of the study (wave 6, 2012–2013). The horizontal axis indicates the respondent's age. The vertical axis indicates the respondent's predicted BMI (figure 1A, B). The figures show that in early old age, there was an increase in both BMI and WC, thereafter BMI and WC decreased

**Table 1** Baseline characteristics of the participants included in the samples used to model trajectories of body mass index (BMI) and waist circumference (WC) over 8 years, the English Longitudinal Study of Ageing 2004–2005

| | BMI sample | | WC sample | |
| --- | --- | --- | --- | --- |
| | n=7225 | | n=7416 | |
| | Mean | SD | Mean | SD |
| Age | 66.2 | 9.4 | 66.3 | 9.5 |
| BMI (kg/m²) | 27.9 | 4.9 | 26.7 | 7.3 |
| WC (cm) | 94.2 | 17 | 95.8 | 13.1 |
| **Wealth quintiles** | **%** | **N** | **%** | **N** |
| Poorest | 22.0 | 1445 | 21.5 | 1484 |
| Second wealth quintile | 20.0 | 1447 | 19.9 | 1483 |
| Third wealth quintile | 20.0 | 1443 | 19.9 | 1483 |
| Fourth wealth quintile | 20.0 | 1445 | 19.7 | 1483 |
| Richest | 19.0 | 1445 | 18.9 | 1483 |
| Female | 53.4 | 3966 | 53.8 | 4079 |
| Cohabiting with a partner | 70.2 | 5039 | 69.4 | 5128 |
| Limiting long-standing illness | 33.8 | 2445 | 34.4 | 2557 |

Means and percentages weighted for non-response.

**Table 2** Latent growth model of anthropometric measures by age, ELSA 2004/2005 to 2012/2013

| | BMI (kg/m²) | | | Waist circumference (cm) | | |
|---|---|---|---|---|---|---|
| | Estimate* | 95% CI | P value | Estimate* | 95% CI | P value |
| Intercept | 28.19 | 28.06 to 28.32 | <0.001 | 96.98 | 96.63 to 97.73 | <0.001 |
| Intercept variance | 23.44 | 22.36 to 24.52 | <0.001 | 158.12 | 152.30 to 163.93 | <0.001 |
| Slope | 0.029 | 0.018 to 0.040 | <0.001 | 0.183 | 0.151 to 0.215 | <0.001 |
| Slope variance | 0.094 | 0.069 to 0.120 | <0.001 | 0.268 | 0.151 to 0.385 | <0.001 |
| Intercept regressed on | | | | | | |
| Baseline age† linear | −0.372 | −0.486 to −0.258 | <0.001 | −0.153 | −0.453 to 0.148 | 0.404 |
| Baseline age† quadratic | −0.276 | −0.378 to −0.174 | <0.001 | −1.042 | −1.300 to −0.780 | <0.001 |
| Slope regressed on | | | | | | |
| Baseline age† linear | −0.061 | −0.072 to −0.051 | <0.001 | −0.086 | −0.114 to −0.059 | <0.001 |
| Baseline age† quadratic | −0.017 | −0.028 to −0.006 | 0.015 | −0.028 | −0.061 to 0.004 | 0.148 |
| Model fit | | | | | | |
| CFI | 0.986 | | | 0.978 | | |
| TLI | 0.959 | | | 0.935 | | |
| RMSEA | 0.096 | | | 0.080 | | |
| N | 7225 | | | 7416 | | |

*Weighted for non-response.
†Baseline age is centred at the grand mean of 66 years and estimates are presented for a 10-year increment.
BMI, body mass index; CFI, Comparative Fit Index; ELSA, English Longitudinal Study of Ageing; RMSEA, root-mean-square error of approximation; TLI, Tucker-Lewis Index, WC, waist circumference.

steeply, from the age of 71 for BMI and from the age of 80 for WC. The graphs also reveal cohort differences in the trajectories of anthropometric measures, so that younger generations (born later) display greater body size indicators compared with older generations. For example, someone born in 1934, who was 70 in 2004 had a BMI of 27.9 kg/m² and a WC of 96.8 cm, whereas someone born in 1942 who was aged 70 in 2012 had both higher BMI and WC (BMI 28.7 kg/m² and WC 98.6 cm).

### Wealth differences in BMI and WC age trajectories

Table 3 shows parameter estimates adjusted for age (linear and quadratic), sex, cohabiting status, and limiting long-standing illness. In the top part of the table, the values 27.00 kg/m² and 100.09 cm represent the estimated baseline mean BMI and WC for a man aged 66 years in the richest wealth quintile with average conditions, which increased 8 years later to 27.12 kg/m² (non-significant change) and 101.19 cm (p<0.001), respectively. For an individual of same age in the poorest quintile of wealth, the mean BMI and WC at baseline were 28.97 kg/m² and 104.75 cm, which increased 8 years later to 29.10 kg/m² and 105.85 cm, respectively. For example, for a man aged 80 years at baseline: his baseline average BMI and WC if he was in the richest wealth quintile were 24.43 kg/

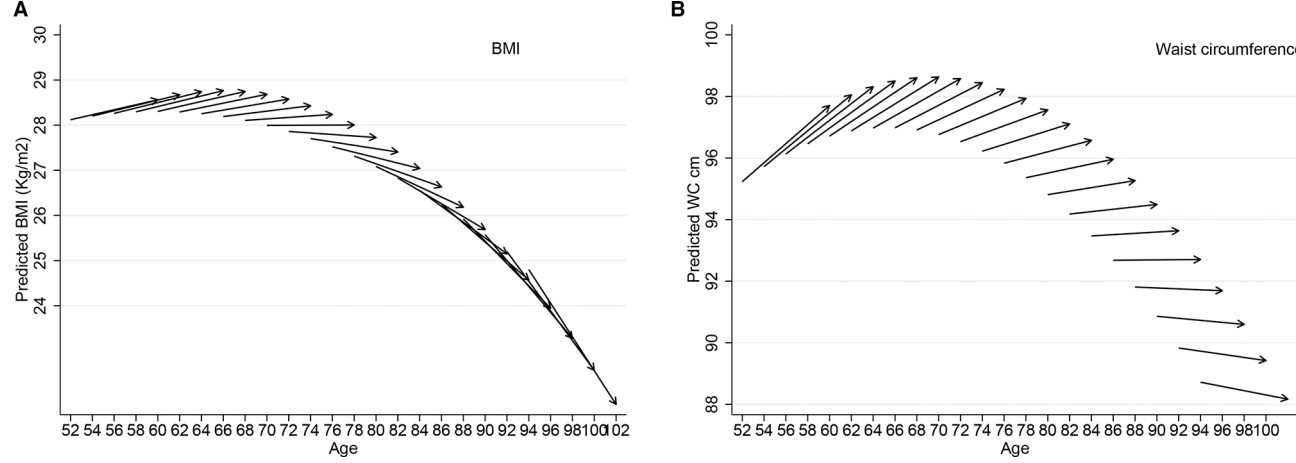

**Figure 1** Vector graph showing 8-year ageing vectors of anthropometric markers, ELSA 2004–2005 to 2012–2013. BMI, body mass index; ELSA, English Longitudinal Study of Ageing; WC, waist circumference.

**Table 3** Adjusted latent growth model of anthropometric measures by age and wealth, ELSA 2004/2005 to 2012/2013

| | BMI (kg/m²) | | | Waist circumference(cm) | | |
|---|---|---|---|---|---|---|
| | Estimate* | 95% CI | P value | Estimate* | 95% CI | P value |
| Intercept | 27.00 | 26.77 to 27.22 | <0.001 | 100.09 | 99.57 to 101.00 | <0.001 |
| Intercept variance | 22.51 | 21.47 to 23.54 | <0.001 | 125.23 | 117.27 to 131.07 | <0.001 |
| Slope | 0.016 | –0.002 to 0.034 | 0.140 | 0.137 | 0.091 to 0.224 | <0.001 |
| Slope variance | 0.094 | 0.069 to 0.118 | <0.001 | 0.315 | 0.206 to 0.450 | <0.001 |
| Intercept regressed on | | | | | | |
| Baseline age† linear | –0.511 | –0.628 to –0.394 | <0.001 | –0.293 | –0.574 to –0.013 | 0.086 |
| Baseline age† quadratic | –0.256 | –0.357 to –0.155 | <0.001 | –0.750 | –0.981 to –0.518 | <0.001 |
| Female | 0.173 | –0.017 to 0.363 | 0.135 | –10.78 | –11.25 to –10.31 | <0.001 |
| Poorest wealth | 1.966 | 0.987 to 1.545 | <0.001 | 4.660 | 3.681 to 5.239 | <0.001 |
| Second wealth quintile | 1.266 | 1.003 to 1.702 | <0.001 | 3.007 | 2.308 to 3.705 | <0.001 |
| Third wealth quintile | 1.458 | 1.181 to 1.735 | <0.001 | 2.916 | 2.229 to 3.603 | <0.010 |
| Fourth wealth quintile | 0.594 | 0.323 to 0.864 | <0.001 | 1.215 | 0.551 to 1.879 | <0.001 |
| Slope regressed on | | | | | | |
| Baseline age* linear | –0.062 | –0.072 to –0.052 | <0.001 | –0.097 | –0.125 to –0.068 | <0.001 |
| Baseline age* quadratic | –0.017 | –0.029 to –0.006 | 0.014 | –0.033 | –0.065 to 0.001 | 0.090 |
| Female | –0.001 | –0.014 to 0.016 | 0.910 | 0.065 | 0.022 to 0.109 | 0.014 |
| Poorest wealth | 0.006 | –0.022 to 0.035 | 0.723 | 0.002 | –0.076 to 0.079 | 0.971 |
| Second wealth quintile | 0.024 | 0.000 to 0.048 | 0.107 | 0.053 | –0.014 to 0.121 | 0.195 |
| Third wealth quintile | 0.030 | –0.008 to 0.052 | 0.027 | 0.070 | –0.008 to 0.133 | 0.064 |
| Fourth wealth quintile | 0.010 | –0.014 to 0.016 | 0.465 | –0.035 | –0.097 to 0.026 | 0.347 |
| Model fit | | | | | | |
| CFI | 0.994 | | | 0.985 | | |
| TLI | 0.983 | | | 0.954 | | |
| RMSEA | 0.020 | | | 0.049 | | |
| N | 7225 | | | 7416 | | |

*Estimates adjusted for marital status, and limiting long-standing illness, all centred to the grand mean.
†Baseline age is centred at the grand mean of 66 years and presented for a 10-year increment. Weighted for non-response.
BMI, body mass index; CFI, Comparative Fit Index; ELSA, English Longitudinal Study of Ageing; RMSEA, root-ean-square error of approximation; TLI, Tucker-Lewis Index.

m² and 98.31 cm, respectively, compared with 26.40 kg/m² and 102.97 cm if he was in the poorest wealth quintile. Eight years later, BMI was 22.34 kg/m² and WC was 96.72 cm for men in the richest wealth quintile compared with 24.31 kg/m² and 101.38 cm in the poorest wealth quintile.

There was no difference in BMI between men and women. However, overall WC at baseline was 10.78 cm lower (p<0.001) in women, but women experienced a significantly higher increase over time compared with men, as indicated by the positive coefficient of sex on the slope (0.065, p<0.05).

Figures 2 and 3 provide summaries of the model presented in table 3, comparing people in the richest and poorest wealth quintiles. All variables including sex (but excluding wealth) have been centred to their overall mean; therefore, the estimates are an average for men and women, with average health and demographic conditions. BMI showed a baseline intercept of 27.09 kg/m² and a change per 4 years of +0.016 kg/m², and WC at baseline was 94.30 cm, with a 4-year change of +0.172 cm.

### Sensitivity analyses
In online supplementary tables 2 and 3, we present the results of the joint model of loss to follow-up with the latent growth curve. Estimates from this model showed that loss to follow-up did not affect estimates of mean levels (intercept and slope) of BMI and WC.

Interaction terms between wealth and sex on the intercepts and slopes of BMI and WC were not significant (online supplementary table 4). Furthermore, the results of multigroup growth curve models showed that trajectories of BMI and WC with wealth do not differ by sex (online supplementary tables 5 and 6), the model fit indicates that this model does not fit the data better than the model with sex as covariate.

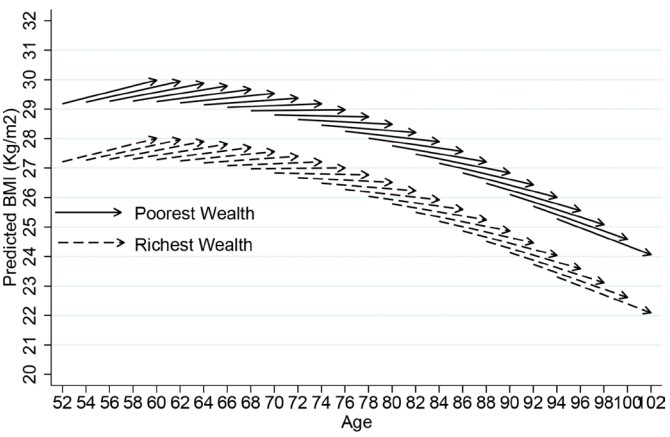

**Figure 2** Vector graph showing adjusted 8-year ageing vectors of BMI by wealth, ELSA 2004–2005 to 2012–2013. BMI, body mass index; ELSA, English Longitudinal Study of Ageing.

## DISCUSSION

Using a large nationally representative sample of older adults in England, we found that age (linear and quadratic) was a strong predictor of both initial levels and changes over time in body size defined by BMI and WC. Our results showed that, in early old age, both BMI and WC increase significantly over the 8-year follow-up, whereas at older ages, both BMI and WC decrease steeply. When the age trajectories were explored by wealth, we found significant differences in initial levels of BMI and WC, with people in the poorest wealth quintile displaying significantly higher BMI and WC than those in the richest quintile. However, the rate of change in both BMI and WC did not differ significantly according to wealth, meaning that the age trajectories of BMI and WC were parallel across wealth categories and that the socioeconomic gap did not close at older ages. At any given age, individuals in the lowest wealth group presented a disadvantage compared with their richer counterparts.

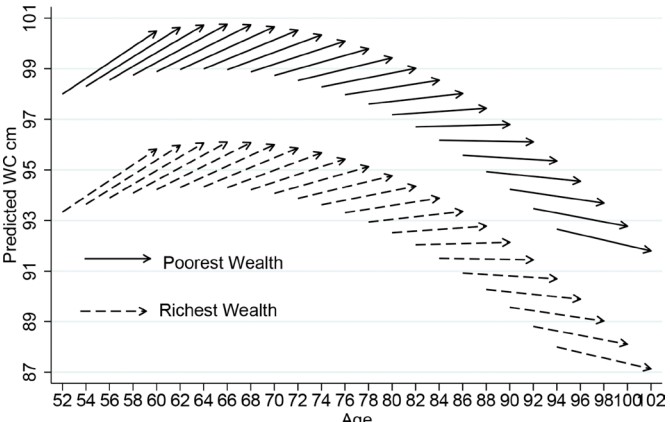

**Figure 3** Vector graph showing adjusted 8-year ageing vectors of WC by wealth, ELSA 2004–2005 to 2012–2013. ELSA, English Longitudinal Study of Ageing; WC, waist circumference.

Longitudinal studies of change in BMI in older ages are limited and yielded mixed results in particular regarding the age at which the trend reverses from positive to negative change over time.[10–12] Overall, our study is consistent with the literature that shows an increase in BMI over time from middle age to early old age and a decrease thereafter. Some studies have found the trend to reverse from age 55,[29 30] whereas, an analysis in the Health and Retirement Study in the USA found the trend reversed from the age of 67 years which is more similar to our finding of a reversed trend from the age of 70.[10]

Accumulation of fat in the visceral region has been shown to be particularly detrimental for cardiometabolic outcomes, as opposed to subcutaneous fat, and WC is a better measure to detect central obesity and overall body fat, compared with BMI which cannot adequately distinguish between fat and lean mass. Despite the potential superiority of this marker for prediction of adverse ageing outcomes, there is only limited evidence of the age-related changes in WC in older adults and how they may vary according to socioeconomic circumstances. We found a significant increase over time in WC in individuals aged up to 79, which is consistent with two longitudinal studies of individuals aged up to 75 years.[31 32] However, our study is the first to describe the decline in WC from the age of 80, as there are no studies that have explored changes in WC among individuals aged over 75 years.

The decrease in WC is delayed by 9 years after the start of decline in BMI. This potentially means that from age 70 onwards, the decrease in BMI reflects a loss of muscle mass not compensated by an increase in fat mass, whereas an age-related change in the fat distribution is reflected by a still increasing WC. A previous study in ELSA showed that underweight is associated with higher risk of frailty, and the more so as in the presence of abdominal obesity.[33] In the older old, from 80 years onwards, we depicted a loss in both lean and fat body mass. An indirect public health implication of our observations is that efforts should be made to emphasise the importance of physical activity and a healthy diet both to limit muscle loss and accumulation of visceral fat at older age.

Socioeconomic differences in trajectories of body size in older adults have been explored using social class, income and/or education. Our study is the first to use wealth as an indicator of socioeconomic position, which is not directly comparable with the existing literature. Nevertheless, our results of a significant association of wealth on baseline levels but not on the rate of change in BMI are in accordance with studies using education in the USA[10] and in a population of older Japanese.[17] However, in the latter study, the trend was opposite when using income, with people with a lower income displaying lower BMI compared with the highest income categories. Comparison with other studies on wealth differences in age trajectories of WC was not possible, since this is the first study to carry out such examination. The measure of wealth we used encompasses assets and net worth, carrying with it information on individuals'

past circumstances, therefore, it might be a better indicator of the permanent socioeconomic status of older adults, compared with income, social class and education. Our results uncover yet another aspect of health social inequalities, whereby disadvantaged populations are more likely to suffer from overweight but also to become frailer at older ages, with inequalities tracking over the life course rather than fading at older ages. This emphasises the need to provide programmes of weight management and lifestyle counselling, as well as frailty assessment in the disadvantaged communities. Research should focus on developing and evaluating the efficacy of such programmes.

This study has several limitations. First, data on BMI and WC before the age of 50 were not collected. Therefore, we were not able to identify the period during the life course when inequalities started. Second, as both overall and central obesity are related to an increased risk of early mortality and chronic conditions, it is possible that we underestimated baseline inequalities in BMI and WC. Another limitation of our study is the proportion of missing data. In order to minimise the potential bias derived from missing data, we used full information maximum likelihood estimation. It is unlikely that attrition substantially influenced our results about wealth disparities in anthropometric measures. We used joint modelling to provide estimates of change in anthropometric measures that allow for correlation between these measures at previous waves and attrition. The results from the joint models (online supplementary tables 1 and 2) were similar, which suggested that the missing at random assumption for missing data was reasonable. Despite these limitations, our study has several strengths that make it unique. It is the first large nationally representative study in England to depict age-specific trajectories of BMI and WC over an 8-year period and to examine whether these differed by wealth. The latent growth curve modelling is an originality of our study compared with the literature, furthermore, both BMI and WC were assessed objectively by a trained nurse as opposed to most studies which used self-reported data. Lastly, we used a measure of socioeconomic status that is particularly relevant at older ages, since it reflects both past and current socioeconomic status.

To conclude, we have shown that older English adults experienced a gain over time in both BMI and WC in early old age but the trend reversed at older old age when body size decreased steeply. People in the poorest wealth groups had overall higher initial levels of BMI and WC than their richer counterparts, however, the change over time in both BMI and WC did not differ according to wealth, meaning that the socioeconomic gap did not close at older ages. Addressing socioeconomic inequalities in total and abdominal obesity and preventing weight gain is a crucial challenge at any stage of life, including at early old age. Equally important is the detection of and prevention of frailty at older old age.

**Acknowledgements** The English Longitudinal Study of Ageing was developed by a team of researchers based at the University College London, NatCen Social Research, the Institute for Fiscal Studies and the University of Manchester. The data were collected by NatCen Social Research.

**Contributors** PZ designed and conducted the study. PZ and CL analysed the data and wrote the manuscript.

**Funding** This work was supported by the National Institute of Aging (R01AG017644) and a consortium of UK government departments coordinated by the Economic and Social Research Council.

**Disclaimer** The funding bodies had no role in the study design; in the collection, analysis and interpretation of data; in the writing of the manuscript; and in the decision to submit the manuscript for publication.

**Competing interests** None declared.

**Patient consent for publication** Not required.

**Ethics approval** ELSA was approved by NHS Research Ethics Committees under the National Research and Ethics Service.

**Provenance and peer review** Not commissioned; externally peer reviewed.

**Data sharing statement** ELSA data are available free on registration to the UK Data Archive http://data-archive.ac.uk/, codes are available directly from the authors.

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
