## [Reviewer comments · BMJ Open]

ARTICLE DETAILS

TITLE (PROVISIONAL)	Socioeconomic trajectories of body mass index and waist circumference: results from the English Longitudinal Study of Ageing
AUTHORS	Zaninotto, Paola; lassale, Camille

VERSION 1 - REVIEW

REVIEWER	Awat Feizi IUMS, Iran
REVIEW RETURNED	26-Aug-2018

GENERAL COMMENTS	Thank you giving me the opportunity for reviewing this submitted manuscript 1.The authors did a well designed and written study on a relatively large samples of adults using an advanced statistical model i.e. latent growth on a longitudinal data for evaluating the trajectories of common anthropocentric measures Please see some minor comments for completing particularly results of study - Abstract: they did not present and information about wealth and used instrument for evaluating it, they stated no relationship between wealth and change in BMI and WC however in conclusion presented some matters about it! or they sated age 52 as a critical point while no data have been presented in results section about that? totally conclusion in results section should be revised extensively -Introduction section: It is strongly recommended to declare the aspects of ability of the used statistical model in this study for more reliable tracking of the changes or time, and in this regards a question: why the authors did not use the other common statistical model for longitudinal data? Methods section: first paragraph should be along with relevant reference to the published studies from ELSA, a bit more details about study participants should be presented, title "Patient and public involvement statement" should be deleted, the instrument used for evaluating the wealth should be introduced more complete and the components and scoring, its reliability and validity should be presented in more details , the "covariate subsection" should be completed through introducing the presented variables in more complete content, please introduce the instrument used for evaluating the PA and its reliability and validity, the sentence "... the presence of a limiting longstanding illness' is not complete and the "longstanding illness" should be introduced. Some theoretical matters about the latent growth
--

	models should be presented focusing on justifying the use of these models for analysis data in current study and the presented information should be based on relevant references , the authors should declare also how the quantitative and categorical data have been presented in results section, the last sentence in methods section does not make sense Results section: more information about the content of table 1 should be presented in the body of results section (first paragraph), the presented data in table 1 needs major revision, the frequency and percentage for categorical data should be presented, the presented 95%CI are irrelevant, title table should be revised!!, the interpretation of latent growth model's results should be reconsidered , please see the published papers in this framework for getting guidance on how to present the results more understandable for clinicians not for expert in Biostatistics and Epidemiology, another important suggestion regarding to completion of results is : please do an stratified analysis by gender and fit the models separately in men and women, -Conclusion:Please consider my point regarding the defect about the conclusion in abstract for this section too
--	---

REVIEWER	Jerome Rachele University of Melbourne, Australia.
REVIEW RETURNED	27-Aug-2018

GENERAL COMMENTS	Thank you for the opportunity to review this manuscript. This study aimed to explore age-trajectories in BMI and waist circumference, and to examine whether these trajectories variable by wealth using a nationally representative cohort study. The study is overall well-written, but I have some queries which I outline below.  - The introduction was well-written and looks to provide a good rationale for the study. - Could the authors provide more information on response rates / attrition? - I think there is a grammar error on line 25 page 5. - Could the authors provide some justification for the inclusion of smoking and physical activity. I would posit them be on the causal pathway between SES and BMI/WC, not confounders, and therefore their inclusion would appear to be over-adjustment. - There needs to be more details on the possible policy implications of this research, in relation to the magnitude of the problem that this research seeks to address, in light of the study findings. So far, the only reference to policy seems to be one sentence on line 37-31 on page 10. - What are the priorities for future research?
--

REVIEWER	Kyoungwoo Kim Inje University, Korea, Republic of
REVIEW RETURNED	31-Aug-2018

GENERAL COMMENTS	1. About ELSA cohort. What was the follow-up loss rate in ELSA cohort during the study waves? Did you use the data with weight adjustment to the representative sample?
--

	2. About wealth measurement, Can you define wealth using other variables? Were total wealth calculated individually, or considering house hold income of husband? Is there a possibility that the change of wealth, not initial wealth status in wave 1, can affect the trajectories differently? How did you divide the wealth into quintiles, in the total population or divide according to every age? Wealth might be closely related with age itself. 3. birth cohort effect, not biological age itself BMI and WC were not same with the same year old people, what I mean for example, WC and BMI of 70 years old in wave 4 were different from those of age 70 years old in wave 1, 8 years ago. 4. gender difference in trajectories The change of BMI and WC might be closely related with the hormonal change velocity, andropause and menopause and trajectories might also different according to wealth of men and women.
--	--

VERSION 1 – AUTHOR RESPONSE

Responses to Reviewers' Comments to Author:

Reviewer: 1

Reviewer Name: Awat Feizi

Institution and Country: IUMS, Iran

Please state any competing interests or state 'None declared': None

Thank you giving me the opportunity for reviewing this submitted manuscript 1. The authors did a well designed and written study on a relatively large samples of adults using an advanced statistical model i.e. latent growth on a longitudinal data for evaluating the trajectories of common anthropometric measures

Response: Thank you for your positive evaluation of our work.

Please see some minor comments for completing particularly results of study

1. Abstract: they did not present and information about wealth and used instrument for evaluating it, they stated no relationship between wealth and change in BMI and WC however in conclusion presented some matters about it! or they stated age 52 as a critical point while no data have been presented in results section about that? totally conclusion in results section should be revised extensively

Response: We agree that the information provided in the abstract deserved clarification. We have now added information about how wealth was defined, as follows: "Main exposure measure: Total non-pension household wealth quintiles defined as financial wealth, physical wealth (such as

business wealth, land, or jewels), and housing wealth (primary and secondary residential housing wealth), minus debts.”

Moreover, we would like to clarify that in the abstract we stated a significant relationship between BMI and wealth and between WC and wealth at baseline but not relationships with rates of change. We have now added some text to clarify this and the Results section now reads:

“Results: Using latent growth curve models we showed that BMI increased by 0.03Kg/m² (95%CI 0.02; 0.04 p<0.001) per year and WC by 0.18cm (0.15,0.22, p<0.001). Age (linear and quadratic) showed a negative association with BMI and WC baseline and rates of change, indicating that older individuals had smaller body sizes and that the positive rates of change flattened to eventually become negative. The decline occurred around the age of 71y for BMI and 80y for WC. Poorest wealth was significantly related to higher baseline levels of BMI (1.97Kg/m² 95%CI:0.99; 1.55, p<0.001) and WC (4.66cm 95%CI:3.68; 2.40, p<0.001). However, no significant difference was found in the rate of change of BMI and WC by wealth, meaning that the age-trajectories of BMI and WC were parallel across wealth categories and that the socioeconomic gap did not close at older ages.”

The conclusions mentioned the starting age of our sample, which is 52 but we understand that this might have created some confusion we have therefore rephrased it. In the light of the new wording, we think that the conclusions are appropriately phrased:

“Conclusions: Older English adults showed an increase in BMI and WC over time but this trend reversed at older old age to display a sharp decrease. At any given age wealthier people had more favourable BMI and WC profile.”

2. Introduction section: It is strongly recommended to declare the aspects of ability of the used statistical model in this study for more reliable tracking of the changes over time, and in this regards a question: why the authors did not use the other common statistical model for longitudinal data?

Response: We would like to clarify that, compared to common statistical models in the context of modelling trajectories over time, growth curve models are the most appropriate and also offer the advantages of higher level of statistical power, analysis at the individual level to examine individual variability in also rates of change, and also parameter estimation with partially missing data.

As suggested by the reviewer we have added the following text at the end of the introduction to address your suggestion:

“To allow reliable tracking of changes over time, we use growth curve models which enable studying between-person differences in within-person change by fitting fixed and random effects”.

3. Methods section: first paragraph should be along with relevant reference to the published studies from ELSA, a bit more details about study participants should be presented.

Response: We have included a reference on the cohort profile of ELSA and have added some additional information about the study, in the first paragraph of the Methods section p.5:

“The data are from three waves of the English Longitudinal Study of Ageing (ELSA),²¹ where the same individuals aged 50 and older and living in private households in England, were followed and re-interviewed every 2 years. The study began in 2002-2003 (first phase of data collection referred to as wave 1) with 11 391 individuals. Data collection includes a face-to-face interview and a self-completion questionnaire at each wave and trained nurses performed visits in the participants’ homes at alternative waves, to collect anthropometry measures and blood samples. For the purpose of this study, we used data from wave 2 (2004-2005), wave 4 (2008-2009) and wave 6 (2012-2013) where nurse visits took place and anthropometric measurements were taken. The sample size consisted of 7255 (5042 at wave 4 and 4256 at wave 6) for the study of BMI trajectories and 7416 (4907 at wave4

and 4147 at wave 6) for the study of waist circumference trajectories. Total follow-up time was 8 years (average 4).”

4. Title "Patient and public involvement statement" should be deleted

Response: There was a misspelling in this sentence, which we have corrected, furthermore this is a requirement of the journal, we have rephrased it as follows:

“Patients did not participate in the design and data analysis of this study.”

5. The instrument used for evaluating the wealth should be introduced more complete and the components and scoring, its reliability and validity should be presented in more details

Response: The aggregate variable of wealth is calculated on 31 components and it is not possible to report them all in detail. To address this comment within the space constraint, we have provided additional text in supplementary materials and added revised the text in the methods section p.6:

“Total non-pension household wealth, reported at the household level, was defined as financial wealth, physical wealth (such as business wealth, land, or jewels), and housing wealth (primary and secondary residential housing wealth), minus debts. The variable is excluding regular pension payments, but includes lump sums from private pension that have already been received but not yet consumed. The total score (range -126990 to 9319227, mean 276702 SD 396453.3) was divided into five equal quintiles. More detailed information can be found in supplementary materials.”

6. The "covariate subsection" should be completed through introducing the presented variables in more complete content,

Response: thank you for your suggestion, we have now included additional information, p.6 under the subheading Covariates:

“We used a continuous variable for age (ranging from 52 to 101 at baseline) and a dichotomous variable for sex (0“Male” 1“Female”). From information on marital and cohabiting status we derived a dichotomous variable for cohabitation status (0“Currently living with a partner whether married or not” 1“Currently not living with a partner”). During the interview respondents were asked whether they had any longstanding illnesses, and if the illnesses limited their daily activities; responses were combined into a dichotomous variable, classifying participants as having a limiting longstanding illness or not.”

7. Please introduce the instrument used for evaluating the PA and its reliability and validity

Response: Following the suggestion of Reviewer 2, we deleted information about smoking and physical activity.

8. The sentence "... the presence of a limiting longstanding illness' is not complete and the "longstanding illness" should be introduced.

Response: Thank you for your comment, we have now rephrased the sentence as follows:

“During the interview respondents were asked whether they had any longstanding illnesses, and if the illnesses limited their daily activities; responses were combined into a dichotomous variable, classifying participants as having a limiting longstanding illness or not.”

9. Some theoretical matters about the latent growth models should be presented focusing on justifying the use of these models for analysis data in current study and the presented information should be based on relevant references ,

Response: We have rephrased that section on page 6 and 7:

“To best depict reliable trajectories of anthropometric measures and to provide insight into how the rate of change relates to other variables, we used latent growth curve modelling.

Growth curve modelling is specifically designed to capture change over time, by estimating latent growth factors for the baseline status (intercept) and rate of change (linear slope) over the 8-year period.²¹⁻²³⁻²⁶ Latent factors representing intercept and slope components were derived from the three observations of anthropometric measures at wave 2 (baseline), wave 4, and wave 6. Factor loadings of the intercept component to all three observations were fixed to 1, and the linear slope component was defined by fixing the parameters to 0 (baseline, wave 2), 4 (wave 4), and 8 (wave 6) so that the slope parameter can be interpreted as the change per year. To depict age trajectories of BMI and WC we fitted models with a linear and a quadratic term of age (non-adjusted). To answer our second research question, we tested the associations of wealth quintiles (5 categories, with ‘richest wealth’ as the reference category) with intercept and slope. The models included age (linear and quadratic, centred at the mean [66y]), cohabiting status (binary, centred at the mean [0.51]), limiting long standing illness (binary, centred at the mean [0.67]) all measured at baseline. For simplicity we only present parameter estimates for age, sex and wealth. To determine the fit of the models, we employed the Comparative Fit Index (CFI), the Tucker–Lewis index (TLI), and the root-mean-square error of approximation (RMSEA), which represents closeness of fit.²⁷ The growth curve models (and descriptive statistics) accounted for the complex survey design and were weighted to adjust for non-response and to make the ELSA sample representative of the population of adults aged 50 and over, living in private households in England. Missing data were handled using full information maximum likelihood estimation, which computes parameter estimates on the basis of all available data under the assumption that data are missing at random. Models were fitted using Mplus version 7.

In order to show graphically the level of BMI and WC at baseline, direction, and amount of change throughout the age range of our sample, we present aging-vector graphs²⁸ of predicted BMI and WC. The ageing graph were fitted using Stata 15.”

The relevant citations added are:

21 Curran PJ, Obeidat K, Losardo D. Twelve Frequently Asked Questions About Growth Curve Modeling. *J Cogn Dev* 2010;11(2):121-36. doi: 10.1080/15248371003699969 [published Online First: 2010/01/01]

23 Muthén BK, ST. Longitudinal studies of achievement growth using latent variable modeling. *Learning and Individual Differences* 1998;10:79-101.

24 Bollen KC, PJ. Latent curve models: A structural equation perspective. Hoboken. NJ: Wiley 2006.

25 Singer JW, JB. Applied longitudinal data analysis: Modeling change and event occurrence. New York: Oxford University Press 2003.

26 Browne MdT, SHC. Models for learning data. In: Collins LH, JL., ed. Best methods for the analysis of change. Washington, DC: American Psychological Association Press 1991:47-78.

10. The authors should declare also how the quantitative and categorical data have been presented in results section,

Response: We have revised the following paragraph to clarify this issue, p.7:

“To answer our second research question, we tested the associations of wealth quintiles (5 categories, with ‘richest wealth’ as the reference category) with intercept and slope. The models included age (linear and quadratic, centred at the mean [66y]), cohabiting status (binary, centred at

the mean [0.51]), limiting long standing illness (binary, centred at the mean [0.67]) all measured at baseline. For simplicity we only present parameter estimates for age, sex and wealth.”

11. the last sentence in methods section does not make sense

Response: We agree that the information provided was not clear and did not add to the previous sentence. We have therefore rephrased it as follows, p.7:

“The ageing graphs were fitted using Stata 15.”

12. Results section: more information about the content of table 1 should be presented in the body of results section (first paragraph),

Response: To address the reviewer’s comment we have rephrased that section as follows:

“Baseline (2004-2005) descriptive characteristics of the samples are shown in Table 1. In the “BMI” analytical sample (n=7225), overall mean age at baseline was 66, and average BMI was 27.9 kg/m², 53% of people were women; 70% were cohabiting with a partner, and 34% reported having a limiting long standing illness. The “Waist Circumference” analytical sample (n=7416) was essentially similar and the average WC was 95.8cm.”

13. the presented data in table 1 needs major revision, the frequency and percentage for categorical data should be presented, the presented 95%CI are irrelevant, title table should be revised!!,

Response: Thank you for this comment. We are now presenting the frequency and percentage of categorical data and have removed the 95%CI. We also have revised the title as follows: “Baseline characteristics of the analytical sample, the English Longitudinal Study of Ageing 2004-2005”

14. the interpretation of latent growth model's results should be reconsidered , please see the published papers in this framework for getting guidance on how to present the results more understandable for clinicians not for expert in Biostatistics and Epidemiology,

Response: We appreciate the reviewer’s point. In revising, we have taken extra care in phrasing the results description and interpretation in a way that is more understandable for clinicians. For example, p.8: “Age was negatively associated both with intercept and slope. The negative estimate of age on the intercept reflects a cohort effect (older generations have lower values of BMI and WC than younger generations). The negative effect of age on the slope means that the overall positive slope observed at age 66 is actually negative at older ages.”

15. another important suggestion regarding to completion of results is : please do an stratified analysis by gender and fit the models separately in men and women,

Response: We agree that it is important to investigate if body size trajectories differ by gender. Therefore, as suggested by the reviewer, we tested for interaction terms between wealth and gender which did not attain statistical significance. We present a table with interaction terms in supplementary materials (Table S4). As a sensitivity analysis, we also report a multigroup growth curve model by gender, which is a growth curve model that allows testing for the fit to the data when performed by the variable of interest, in this case gender. We added the following text in the results:

“Interaction terms between wealth and gender on the intercepts and slopes of BMI and WC were not significant (Supplementary Table 4). Furthermore, the results of multi-group growth curve models showed that trajectories of BMI and WC with wealth do not differ by gender (supplementary Tables 5 and 6), the model fit indicates that this model does not fit the data better than the model with gender as covariate.”

16. Conclusion: Please consider my point regarding the defect about the conclusion in abstract for this section too

Response: We have now rephrased the conclusions as follows:

“To conclude, we have shown that older English adults experienced a gain over time in both BMI and WC in early old age but the trend reversed at older old age when body size decreased steeply. People in the poorest wealth groups had overall higher initial levels of BMI and WC than their richer counterparts, however, the change over time in both BMI and WC did not differ according to wealth, meaning that the socioeconomic gap did not close at older ages. Addressing socioeconomic inequalities in total and abdominal obesity and preventing weight gain is a crucial challenge at any stage of life, including at early old age. Equally important is the detection of and prevention of frailty at older old age.”

Reviewer: 2

Reviewer Name: Jerome Rachele

Institution and Country: University of Melbourne, Australia.

Please state any competing interests or state 'None declared': None declared

Thank you for the opportunity to review this manuscript. This study aimed to explore age-trajectories in BMI and waist circumference, and to examine whether these trajectories variable by wealth using a nationally representative cohort study. The study is overall well-written, but I have some queries which I outline below.

- The introduction was well-written and looks to provide a good rationale for the study.

Response: Thank you very much for your positive comment

- Could the authors provide more information on response rates / attrition?

Response: Thank you for raising this important point. We are now providing a table with attrition rates in Supplementary materials as Table S1.

- I think there is a grammar error on line 25 page 5.

Response: Thank you for spotting this error, we have now rectified it.

- Could the authors provide some justification for the inclusion of smoking and physical activity. I would posit them be on the causal pathway between SES and BMI/WC, not confounders, and therefore their inclusion would appear to be over-adjustment.

Response: Thank you very much for this important point. We agree with the reviewer that inclusion of smoking and physical activity might result in over-adjustment. After excluding them from the models, the results remain similar. Therefore, following the reviewer's advice, for ease of interpretation and to avoid the risk of over-adjustment, we now present the results without these variables.

- There needs to be more details on the possible policy implications of this research, in relation to the magnitude of the problem that this research seeks to address, in light of the study findings. So far, the only reference to policy seems to be one sentence on line 37-31 on page 10.

Response: Thank you for this comment. We have now expanded on the social inequality aspect and added the following paragraph p.12 : “Our results uncover yet another aspect of health social

inequalities, whereby disadvantaged populations are more likely to suffer from overweight but also to become frailer at older ages, with inequalities tracking over the life course rather than fading at older ages. This emphasises the need to provide programmes of weight management and lifestyle counselling, as well as frailty assessment in the disadvantaged communities.

- What are the priorities for future research?

Response: Thank you for an important question. We believe that designing intervention programmes that aim at reducing social inequalities in health, starting with weight management, should be the priority for future research. We have added the following sentence on p.12 "Research should focus on developing and evaluating the efficacy of such programmes."

Reviewer: 3

Reviewer Name: Kyoungwoo Kim

Institution and Country: Inje University, Korea, Republic of

Please state any competing interests or state 'None declared': None declared

1. About ELSA cohort.

1.1. What was the follow-up loss rate in ELSA cohort during the study waves?

Response: We have added the information on follow-up loss in a Supplemental table Table S1. We now describe it in the Method section p. 5:

"The sample size consisted of 7255 (5042 at wave 4 and 4256 at wave 6) for the study of BMI trajectories and 7416 (4907 at wave4 and 4147 at wave 6) for the study of waist circumference trajectories. Attrition rates are presented in Supplementary Table S1. Total follow-up time was 8 years (average 4)."

1.2. Did you use the data with weight adjustment to the representative sample?

Response: We have used weights to adjust the sample to be representative of the English population of adults aged 50 and over. We have added the following sentence to clarify, p.7:

"The growth curve models (and descriptive statistics) accounted for the complex survey design and were weighted to adjust for non-response and to make the ELSA sample representative of the population of adults aged 50 and over, living in private households in England."

2. About wealth measurement,

Can you define wealth using other variables? Were total wealth calculated individually, or considering house hold income of husband?

Response: Wealth is computed using several variables, such as primary and secondary housing, jewellery, pieces of arts and antiques, etc.. it is a composite measure and it calculated at the household level and then recoded into benefit units. We have added further information as supplementary materials about the definition of the variable and the following text in the methods:

"Total non-pension household wealth, reported at the household level, was defined as financial wealth, physical wealth (such as business wealth, land, or jewels), and housing wealth (primary and secondary residential housing wealth), minus debts. The variable is excluding regular pension payments, but includes lump sums from private pension that have already been received but not yet

consumed. The total score (range -126990 to 9319227, mean 276702 SD 396453.3) was divided into five equal quintiles. More detailed information can be found in supplementary materials.”

3. Is there a possibility that the change of wealth, not initial wealth status in wave 1, can affect the trajectories differently?

Response: We agree that change of wealth could potentially affect the trajectories differently but we argue that wealth is carrying with it information on individuals' past circumstances, and is a good indicator of the permanent socioeconomic position of older adults, and we found that it is less likely to change over an 8-year period than income or occupation. Modelling the change of wealth in parallel with the change of BMI / WC is feasible but beyond the scope of this work and would complicate the interpretation of the results. Furthermore, given that the majority of the sample remains in the same categories of wealth between the first and the last assessments, we think that this set of analysis would not provide additional information.

4. How did you divide the wealth into quintiles, in the total population or divide according to every age? Wealth might be closely related with age itself.

Response: the categories were computed based on the analytical sample used in the study at baseline and not by age. The reason is that wealth, unlike income, is more stable at older ages, since it excludes pension and work-related income, and is based on more durable assets accumulated over the life course.

5. Birth cohort effect, not biological age itself. BMI and WC were not same with the same year old people, what I mean for example, WC and BMI of 70 years old in wave 4 were different from those of age 70 years old in wave 1, 8 years ago.

Response: That is correct and it is a good point to mention. We have clarified this in various instances in the results page 8:

“Age was negatively associated both with intercept and slope. The negative estimate of age on the intercept reflects a cohort effect (older generations have lower values of BMI and WC than younger generations). The negative effect of age on the slope means that the overall positive slope observed at age 66 is actually negative at older ages.”

And p. 9: “The graphs also reveal cohort differences in the trajectories of anthropometric measures, so that younger generations report higher values compared older generations. For example, someone aged 70 in 2004 had BMI 27.9 Kg/m² and WC 96.8cm, whereas someone aged 70 in 2012 had BMI 28.7 Kg/m² and WC 98.6cm.”

5. gender difference in trajectories

The change of BMI and WC might be closely related with the hormonal change velocity, andropause and menopause and trajectories might also differ according to wealth of men and women.

Response: This is an important point and we have further explored whether relationships of BMI and WC with wealth might differ by gender. We found that the interaction terms between gender and wealth on the slopes of BMI and WC were not statistically significant (See supplementary table S4). We also present the results of a multigroup model by gender as sensitivity analysis in supplementary tables 5 and 6. We have added the following text in the results section:

“Interaction terms between wealth and gender on the intercepts and slopes of BMI and WC were not significant (Supplementary Table 4). Furthermore, the results of multi-group growth curve models showed that trajectories of BMI and WC with wealth do not differ by gender (supplementary Tables 5

and 6), the model fit indicates that this model does not fit the data better than the model with gender as covariate.”

VERSION 2 – REVIEW

REVIEWER	Awat Feizi IUMS
REVIEW RETURNED	21-Dec-2018

GENERAL COMMENTS	Dear authors Thanks Majority of comments have been addressed sufficiently; however; apologizes; the following are remained; 7,10 and 13 from first round and the next 2 are new 7. Please introduce the instrument used for evaluating the PA and its reliability and validity Response: Following the suggestion of Reviewer 2, we deleted information about smoking and physical activity. 10. The authors should declare also how the quantitative and categorical data have been presented in results section, 13. title table should be revised!!, 1. The presented results in last row in table 2 is not correct; a confidence interval -0.001;-0.000??!!(negative zero?!!) is significant at p=0.015 but a point estimate 0?!! With confidence interval -0.001;0.000? is not significant (p=0.148) all presented results in this row need major concern please check other results in this table and 3 2. In some presented confidence intervals I do not see symmetry around the zero?!!
---

REVIEWER	Jerome Rachele The University of Melbourne, Australia
REVIEW RETURNED	29-Nov-2018

GENERAL COMMENTS	The authors have appropriately responded to each of my comments and made suitable revisions to the manuscript. I have no further comments.
--

REVIEWER	Kyoungwoo KIm Inje University, Republic of Korea
REVIEW RETURNED	10-Dec-2018

GENERAL COMMENTS	Revised manuscpit seems to reflect my review opinions
---

VERSION 2 – AUTHOR RESPONSE

Responses to Reviewer_1's Comments to Author:

Reviewer: 1

Reviewer Name: Awat Feizi

Institution and Country: IUMS, Iran

Please state any competing interests or state 'None declared': None

Dear authors Thanks Majority of comments have been addressed sufficiently; however; apologizes; the following are remained; 7,10 and 13 from first round and the next 2 are new

7. Please introduce the instrument used for evaluating the PA and its reliability and validity

Response: We are sorry if our response was not clear in the revision submitted. We assume that by PA the reviewer refers to physical activity. If that is the case, this variable is no longer used in the present study as suggested in revision 1 by reviewer 2, that is the reason for not describing the instrument used for its evaluation.

10. The authors should declare also how the quantitative and categorical data have been presented in results section:

Response: we have added the following text in the manuscript page 7:

“To answer our second research question, we tested the associations of wealth quintiles (treated as categorical in the models with ‘richest wealth’ as the reference category) with intercept and slope. All models included age (linear and quadratic, centred at the mean [66y]), sex (treated as binary), cohabiting status (treated as binary, centred at the mean [0.51]), limiting long standing illness (treated as binary, centred at the mean [0.67]) as covariates, all measured at baseline. To ease interpretation of estimates for age (linear and quadratic) we used a 10 years increment in age, instead of single years.”

13. title table should be revised!!

Response: in revision 1 we did revise the title, but we assume that the referee was not satisfied, therefore we made a further revision, which we hope addresses this comment. The title now reads:

“Baseline characteristics of the participants included in the samples used to model trajectories of body mass index (BMI) and waist circumference (WC) over 8 years, the English Longitudinal Study of Ageing 2004-2005”

New comments

1. The presented results in last row in table 2 is not correct; a confidence interval -0.001;-0.000?!!(negative zero?!!) is significant at $p=0.015$ but a point estimate 0?!! With confidence interval -0.001;0.000? is not significant ($p=0.148$) all presented results in this row need major concern please check other results in this table and 3

2. In some presented confidence intervals I do not see symmetry around the zero?!!

Response: We understand the reviewer's concern, however, these results are correct. The reasons for these confidence being very close to 0, or 0, is due to the fact that age is entered into the model as continuous with 1 year increment, this reflects the small precision of the estimates, more decimal points would be needed to present the actual value of the confidence intervals.

In order to address this comment, we now present the estimates per 10 years increase in age, so that the estimates and related confidence intervals show increased precision and are easier to interpret.

VERSION 3 - REVIEW

REVIEWER	Awat Feizi IUMS
REVIEW RETURNED	05-Feb-2019

GENERAL COMMENTS	Dear editor The last requested revisions have been addressed by author; it is acceptable.
--